# Multiple Selection Criteria for Probiotic Strains with High Potential for Obesity Management

**DOI:** 10.3390/nu13030713

**Published:** 2021-02-24

**Authors:** Jeanne Alard, Benoit Cudennec, Denise Boutillier, Véronique Peucelle, Amandine Descat, Raphaël Decoin, Sarah Kuylle, Amin Jablaoui, Moez Rhimi, Isabelle Wolowczuk, Bruno Pot, Anne Tailleux, Emmanuelle Maguin, Sophie Holowacz, Corinne Grangette

**Affiliations:** 1U1019—UMR 9017—CIIL—Centre d’Infection et d’Immunité de Lille, Institut Pasteur de Lille, Université de Lille, CNRS, Inserm, CHU Lille, F-59000 Lille, France; Jeanne.alard@gmail.com (J.A.); denise.boutillier@orange.fr (D.B.); veronique.peucelle@pasteur-lille.fr (V.P.); isabelle.wolowczuk@pasteur-lille.fr (I.W.); bruno.pot@vub.ac.be (B.P.); 2Université de Lille, UMR-T 1158, BioEcoAgro, F-59000 Lille, France; benoit.cudennec@univ-lille.fr; 3EA 7365-GRITA-Groupe de Recherche sur les formes Injectables et les Technologies Associées, Université de Lille, CHU Lille, F-59000 Lille, France; amandine.descat@univ-lille.fr; 4Institut Pasteur de Lille, Université deLille, Inserm, CHU Lille, U1011—EGID, F-59000 Lille, France; raphael.decoin.etu@univ-lille.fr (R.D.); anne.tailleux@univ-lille.fr (A.T.); 5GENIBIO, Le Pradas, ZI du Couserans, 09190 Lorp-Sentaraille, France; sarah.kuylle@gmail.com; 6Institut Micalis, INRAE, AgroParisTech, Université Paris-Saclay, F-78350 Jouy-en-Josas, France; amin.jablaoui@gmail.com (A.J.); moez.rhimi@inrae.fr (M.R.); emmanuelle.maguin@inrae.fr (E.M.); 7PiLeJe Laboratoire, 37 Quai de Grenelle, 75015 Paris, France; s.holowacz@pileje.com

**Keywords:** probiotics, microbiota, obesity, inflammation, epithelial barrier, enteroendocrine peptides, hypothalamus, bile acids

## Abstract

Since alterations of the gut microbiota have been shown to play a major role in obesity, probiotics have attracted attention. Our aim was to identify probiotic candidates for the management of obesity using a combination of in vitro and in vivo approaches. We evaluated in vitro the ability of 23 strains to limit lipid accumulation in adipocytes and to enhance the secretion of satiety-promoting gut peptide in enteroendocrine cells. Following the in vitro screening, selected strains were further investigated in vivo, single, or as mixtures, using a murine model of diet-induced obesity. Strain *Bifidobacterium longum* PI10 administrated alone and the mixture of *B. animalis* subsp. *lactis* LA804 and *Lactobacillus gasseri* LA806 limited body weight gain and reduced obesity-associated metabolic dysfunction and inflammation. These protective effects were associated with changes in the hypothalamic gene expression of leptin and leptin receptor as well as with changes in the composition of gut microbiota and the profile of bile acids. This study provides crucial clues to identify new potential probiotics as effective therapeutic approaches in the management of obesity, while also providing some insights into their mechanisms of action.

## 1. Introduction

The incidence of non-communicable chronic diseases has increased dramatically in recent decades. Obesity, characterized by an excessive body fat accumulation, which is a consequence of an energy imbalance between intake and expenditure, has become a global public health problem. Visceral adiposity results from adipocyte hyperplasia and/or hypertrophy, and is correlated with the accumulation of pro-inflammatory macrophages that release inflammatory cytokines, triggering altered insulin signaling and sensitivity, which are key factors in determining the risk for the development of metabolic diseases, notably Type 2 diabetes mellitus (T2DM) [1].

The gut microbiota has been shown to be an important factor in the pathogenesis of obesity. Compared to conventional mice, germ-free (GF) mice have reduced adiposity, which can be reversed by colonization with a normal gut microbiota [2]. Fecal transplantation from obese donor mice to GF recipients results in a greater increase in body fat in comparison to colonization with a lean microbiota [3]. Similar findings were observed upon transfer of fecal samples from twins discordant for obesity [4]. The microbiota of obese humans as well as that of rodents shows an altered microbial composition. The most frequently cited changes in the microbiome of obese people compared to lean individuals include a drop in bacterial diversity, and a shift in the proportion of the two main dominant phyla Firmicutes and Bacteroidetes [3], even though data on this observation are still questioned [5]. Obesity is also linked to chronic low-grade inflammation, and a disruption in the integrity of the intestinal barrier, notably characterized by a reduction of tight junction proteins and mucus production. This increased permeability may allow the translocation of pro-inflammatory bacterial compounds, such as lipopolysaccharides (LPS), responsible for metabolic endotoxemia and leading gradually to chronic low-grade inflammation and insulin resistance [6]. Conversely, there is growing evidence that short chain fatty acids (SCFAs), derived from bacterial fermentation, play a key role in enteroendocrine regulation, notably by inducing the release of intestinal peptides, such as glucagon-like peptide-1 (GLP-1), which influence satiety and improve glucose homeostasis by activating the G-protein coupled receptors (GPRs), GRP41 and GRP43 [7]. SCFAs have also been shown to serve as a source of energy for colonocytes, reducing permeability and increasing mucus production [8], thereby, improving gut barrier function. Another important class of metabolites, bile acids synthesized in the liver from cholesterol and metabolized by the gut microbiota, can regulate several host functions, including energy metabolism [9]. It has been shown that the limited efficacy of current dietary treatments and the poor success in maintaining body weight loss are related to persistent microbiome alterations contributing to metabolic alterations [10]. Therefore, in addition to lifestyle modifications, the manipulation of gut microbiota could represent a valuable therapeutic option in the fight against obesity.

The administration of probiotics, defined as “live microorganisms that, when administered in adequate amounts, confer a health benefit on the host”, have been proposed as an interesting strategy in the management of obesity [11]. We and others [12,13,14,15,16] have reported the potent beneficial effects of several strains or mixtures of mainly bifidobacteria or lactobacilli in murine models of obesity, showing an improvement in immune and metabolic parameters, including the restoration of insulin sensitivity [11,13,16]. We correlated the positive effect of a specific probiotic mixture on an impact on the composition of the gut microbiota, including restoration of *Akkermansia muciniphila* and *Rikenellaceae* levels [12]. Conversely, we observed an increase in the abundance of *Lactobacillaceae* in obese mice, which was reversed by the administration of a probiotic cocktail containing a *Lactobacillus* strain [12]. Similar findings have been reported in obese and diabetic patients, with higher fecal *Lactobacillus* abundances, correlated with higher blood glucose levels [16,17,18]. It is not yet known whether lactobacilli contribute to T2DM or whether their increase is a consequence of the disease, since different *Lactobacillus* strains have been shown to improve glucose intolerance in mice. It, therefore, remains important to better decipher the potential impact of probiotics. Probiotics may exert their beneficial properties by different mechanisms, including immunomodulation abilities [19], strengthening of the intestinal barrier [20,21], regulation of lipid metabolism, and induction of entero-endocrine regulatory pathways [11].

In a previous study, we have screened 23 strains from the PiLeJe collection, including 15 lactobacilli and six bifidobacteria of different species, for their immunomodulatory properties and their ability to restore the integrity of the gut barrier [20]. The strains with the best anti-inflammatory profiles were mainly bifidobacteria, as shown by their high IL-10/IL12 ratio (Appendix A). Only *B. animalis* subsp. *lactis* LA306 induced a moderate level of IL-10. The immunomodulatory profile of lactobacilli was strain-dependent. The highest anti-inflammatory profile was observed for *Li. salivarius* LA307, *La. rhamnosus* LA305, and *L. casei* PI20 [20]. The ability of the strains to restore and/or strengthen the gut barrier integrity has been evaluated using the polarized Caco-2 cell monolayer epithelial barrier model, sensitized with hydrogen peroxide (H_2_O_2_). All strains were able to decrease the H_2_O_2_-induced drop of the trans-epithelial electrical resistance (TEER), except *Lc. lctis* PI23 and *S. thermophilus* PI21. Some strains, in particular *L. helveticus* PI5, *L. acidophilus* PI11, *Bifidobacterium animalis* subsp. *lactis* LA 804, and *L. gasseri* LA806, were able to restore and even reinforce the epithelial barrier [20].

The objective of the present work was to extend this study by evaluating the ability of the strains to induce the release of enteroendocrine peptides and to limit the accumulation of lipids in adipocytes in vitro. Based on all the in vitro screening tests, we selected the most potent strains to be evaluated for their protective effect in a diet-induced obesity mouse model. Our results demonstrated that three of the selected strains had strong positive effects, alone or in combination.

## 2. Materials and Methods

### 2.1. Bacterial Strains and Culture Conditions

Twenty-three bacterial strains were evaluated in this study, including six bifidobacteria, 15 lactobacilli, one *Streptococcus thermophilus*, and one *Lactococcus lactis* provided by PiLeJe Laboratoire (Table 1). Lactobacilli were grown at 37 °C in De Man, Rogosa, and Sharpe (MRS) broth (Difco, Detroit, MI, USA) and bifidobacteria were cultured at 37 °C in anaerobic conditions (GENbag anaer, bioMérieux, Lyon, France), in de Man, Rogosa and Sharpe (MRS) medium supplemented with 0.1% (*w*/*v*) L-cysteine hydrochloride (Sigma, St-Louis, MO, USA). *Lc. lactis* and *S. thermophilus* were grown in M17 broth (Difco, Detroit, MI, USA) supplemented with 0.5% glucose, respectively, at 30 °C and 42 °C. After overnight culture, bacteria were washed twice in sterile phosphate-buffered saline (PBS, pH 7.2) and resuspended in PBS (final concentration of 2 × 10^9^ colony forming units (CFU)/mL) for in vitro studies. For in vivo administration, lyophilized powders (provided by PiLeJe Laboratoire) were resuspended in PBS at 1 × 10^9^ CFU per mice in 30 µL.

### 2.2. Enteroendocrine Cell Culture and Impact of Probiotics on Gut Peptide Secretion

The murine intestinal secretin tumor cell line STC-1 [22] was grown in Dulbecco’s Modified Eagle’s medium (DMEM) (Life Technologies, Foster City, CA, USA), supplemented with 10% foetal calf serum (Dutscher, Brumath, France), 5 mM of L-glutamine, and 100 µg/mL streptomycin and penicillin, at 37 °C in a 5% CO_2_-95% air atmosphere. For bacterial stimulation, STC-1 cells were grown in 24-well plates at 40,000 cells/well for 72 h. Cells were washed twice with PBS and resuspended in 400 µL of HEPES 20 mM/Tris 20 mM pH 7.4 buffer containing NaCl 140 mM, KCl 4.5 mM, CaCl_2_ 1.2 mM, MgCl_2_ 1.2 mM, and glucose 10 mM, and stimulated with bacteria at a MOI 10:1, for 8 h at 37 °C in a 5% CO_2_-95% air atmosphere. The resulting supernatants were centrifuged at 8000× *g* for 10 min and stored at −20 °C. Quantification of active GLP-1 was performed by Radio-Immuno Assay (RIA) using the EMD Millipore assay (Billerica, MA, USA).

### 2.3. Differentiation of 3T3-L1 Pre-Adipocytes into Adipocytes and Impact of Probiotics on Lipid Accumulation

The 3T3-L1 pre-adipocyte (mouse) cell line (kindly provided by Dr. Sophie Lestavel, Inserm U1011-EGID, Lille, France) was grown in DMEM medium (Life Technologies) supplemented with 10% foetal calf serum (Gibco, Life Technologies, Grand Island, NY, USA), 5 mM of L-glutamine, and 100 µg/mL streptomycin and penicillin, at 37 °C in 5% CO_2_-95% air atmosphere. To differentiate into adipocytes, cells were seeded at 3500 cells/well (12-well plates) in complete DMEM medium for 48 h and further grown in complete DMEM medium supplemented with 0.25 µM dexamethasone, 0.5 mM 3-iso-butyl-1-methylxanthine, and 1 µg/mL insulin (Sigma-Aldrich, St. Louis, MO, USA) for 48 h. Cells were then grown for two more days in complete DMEM medium containing 1 µg/mL insulin before being placed into complete DMEM medium for an additional four days. After these 10 days of differentiation, 70% to 80% of the cells showed lipid droplets under a microscope. The differentiated cells were treated with the bacteria at a ratio of 10:1, bacteria to cell, for 24 h in the presence of gentamycin (3 µg/mL, Gibco) in complete DMEM medium.

The intracellular accumulation of lipids was quantified using Oil-red staining. Briefly, cells were washed twice with PBS and stained using 1 mL of ready-to-use Oil-red-O solution (DiaPath, Martinengo, Italy) for 15 min at room temperature. After three washings with PBS, the dye was eluted for 30 min with 100% isopropanol. Three absorbances (OD 490 nm) were measured by a spectrophotometer (EL_X_808, BioTek Instruments, Winooski, VT, USA). The percentage of stained cells relative to control cells without bacteria was calculated as: (A490 nm [probiotic sample]/A490 nm [control]) × 100.

### 2.4. Murine Model of Diet-Induced Obesity

C57BL/6 JRj male mice (5-week-old) were purchased from Janvier Labs (Le Genest-St-Isle, France). Mice were housed under specific pathogen-free condition in the animal facility of the Institut Pasteur de Lille (accredited no. A59107) and maintained in a temperature-controlled (20 ± 2 °C) facility with a strict 12-h dark/light cycle. Housing and procedures complied with current national and institutional regulations and ethical guidelines (Institut Pasteur de Lille/B59-350009 and CEEA 75 Nord-Pas-de-Calais). Animal experiments were approved by the Ministère de l’Education Nationale, de l’Enseignement Supérieur et de la Recherche, France (accreditation no. APAFIS#2019101811141602). After a one-week acclimation period, animals were given *ad libitum* access to regular chow and water. High-fat diet (HFD, 45% kcal fat, D12451) and control, (LFD, 10% kcal fat, D12450B), irradiated diets were purchased from Research Diets (Brogaarden, Lynge, Denmark). The composition of the diets is detailed in Appendix A.

Mice were randomly assigned and received a daily oral administration of 10^9^ colony-forming units (CFU) of the bacteria resuspended in sterile phosphate-buffered saline (PBS) or PBS alone for control low fat diet (LFD) and high fat diet (HFD) mice, five consecutive days per week. For bacterial mixtures, mice were administered consecutively with 5 × 10^8^ CFU of each strain. After one week, treated mice or control HFD mice were fed with an HFD diet (*n* = 8 per group) and control LFD mice (*n* = 6) were fed with an LFD diet, for an additional 14 weeks, while continuing to receive the bacteria five days per week. Body weight and food intake were weekly recorded. The intraperitoneal glucose tolerance (IP-GTT) test was performed as previously described [12] after 12 weeks of dieting. Portal vein blood, subcutaneous (inguinal) (SCAT), visceral (epididymal) (EWAT) adipose tissues, intestinal segments, and cecal contents were collected at sacrifice. Plasma levels of leptin were measured by enzyme-linked immunosorbent assay (ELISA) using a specific kit (R&D Systems, Minneapolis, MN, USA).

### 2.5. RNA Extraction and Gene Expression Analysis

Immediately after sacrifice, tissue fragments were put in RNAlater^®^ (Ambion, Life Technologies) and frozen at −80 °C. Tissue samples were homogenized using Lysing Matrix D (MP Bio, Eschwege, Germany). Total RNA was purified using a Macherey-Nagel NucleoSpin RNAII isolation kit (Duren, Germany), according to the manufacturer’s recommendation. The quantity and quality of RNA were checked using a NanoDrop spectrophotometer (260/280 nm ratio higher than two in all samples). RNA (1 µg) was reverse-transcribed using the high-capacity cDNA reverse transcription kit (Applied Biosystems^TM^, Warrington, UK). Real-time quantitative PCR (RT-qPCR) was performed using the Power SYBR Green PCR Master Mix on the QuantStudio™ 12K Flex Real-Time PCR System (Applied Biosystems, Warrington, UK). Primers used in the study are indicated in Appendix A. The 2^−ΔΔCT^ method was used to normalize gene expression. 

### 2.6. Analysis of the Cecal Microbiota Composition

Pyrosequencing of cecal bacterial 16S rRNA gene.

The V3-V4 region was amplified from purified genomic DNA, as previously described [12]. The sequenced samples were analyzed using the bioinformatics pipeline FROGS (Find Rapidly OTU with Galaxy Solution) [23]. Using Flash, we assembled the sequences trimmed for adaptors [24]. PCR primers were removed and sequences with sequencing errors in the primers were eliminated (cutadapt) [25]. For each sample, chimera were removed using Vsearch and Uchime [26,27]. Swarm allowed to cluster reads into Operational Taxonomic Units (OTUs) using a threshold at a Reads 97% identity level [28,29]. A reference sequence was then selected for each OTU for assignation at taxonomic levels from the Kingdom to species using the Silva database [30] and the Ribosomal Database Project (RDP) classifier [31].

After rarefaction to even sequencing depths of all microbiome samples, PERMANOVA (permutational multivariate analysis of variances) allowed us to assess the interventions’ effect on microbiota compositions. Significance was checked using Adonis statistical tests (vegan package of R) to evaluate the distances at 9999 permutations between groups. The abundances of given microbial families were calculated by gathering all OTUs belonging to these families. At phylum, family, and genus levels, the Kruskal-Wallis test followed by Dunn’s post-hoc test were performed to evaluate abundance differences between the studied groups. Benjamini-Hochberg corrections (BH) were used to avoid false positives (significance threshold = 0.05) [32]. Data with *p* values ≤ 0.05 were considered to be significant. The R software was used for all statistical analyses (R Core Team, 2015).

### 2.7. Determination of Short Chain Fatty Acids (SCFA) Concentrations

The concentrations of SCFAs in the cecal content were evaluated after extraction with diethyl ether using a GC-2014 gas chromatograph (Shimadzu, Hertogenbosch, The Netherlands) by the ProDigest company, as described [33]. Results were expressed as mmol/g of cecal content.

### 2.8. Bile Acid (BA) Analysis

Plasma concentrations of the bile acids (BAs) were determined after protein precipitation with iced methanol as previously published [34]. BAs from the cecal content were quantified after extraction on samples lyophilized at −80 °C to avoid bacterial BA. In both biological samples, the 27 BA species were quantified by high-performance liquid chromatography (UFLC-XR device, Shimadzu, Kyoto, Japan) coupled to tandem mass spectrometry (MS/MS) (QTRAP 5500 hybrid system, equipped with a Turbo VTM ion source, Sciex, Foster City, CA, USA) using five deuterated BAs (D4-CA, D4-GCA, D4-TCA, D4-CDCA, D4-GCDCA) as internal standards. Plasma and cecal content BA concentrations were expressed in nmol/l and as a percentage of total BAs, respectively. Ratio and total values were determined according to formulas presented in Appendix A.

### 2.9. Statistical Analysis

Graph preparation and statistical evaluation were performed using GraphPad Prism. Differences between groups were assessed using ANOVA, followed by a non-parametric Mann-Whitney test. Differences with *p* value ≤ 0.05 were considered significant.

## 3. Results

### 3.1. Ability of the Strains to Limit Lipid Accumulation in Adipocytes

The impact of bacterial strains on the accumulation of lipids in mature adipocytes was analyzed using differentiated 3T3-L1 murine cells and Oil-red-O staining. As shown in Figure 1A, among the 23 tested strains, 14 were able to limit adipocyte lipid accumulation. However, compared to untreated cells, the effect was statistically significant for four strains only: *Li. salivarius* LA 307, *B. longum* PI10, *Li. salivarius* PI2, and *Lc. lactis* PI23 (*p* < 0.05 or 0.01). In contrast, seven probiotic strains had a tendency to increase lipid accumulation in adipocytes, yet not significantly. The impact of *B. longum* PI10 and *L. salivarius* PI2 on adipocyte lipid accumulation was further confirmed by the increased expression of genes encoding key regulators of adipose lipid metabolism: the peroxisome proliferator-activated receptor (PPAR) gamma (*pparg*) and the lipoprotein lipase (*lpl*) in treated adipocytes, compared to untreated cells (Figure 1B).

### 3.2. Ability of the Strains to Induce the Secretion of GLP-1 by Enteroendocrine Cells

To evaluate the ability of strains to stimulate the secretion of gut peptides, we used the STC-1 murine cell line, which exhibited a phenotype of intestinal endocrine L-cells capable of secreting GLP-1 [35]. As shown in Figure 2, all tested strains were able to enhance the release of GLP-1, with a marked strain-dependent effect. Only 13 strains out of the 23 induced a significant increase in GLP-1 production: *L. acidophilus* PI11, *B. bifidum* LA803, *La. casei* PI20, *B. animalis* subsp. *lactis* LA804, *B. breve* LA 805, *Lp. plantarum* PI3, *Lp. plantarum* PI19, *B. animalis* subsp. *lactis* LA306, *L. helveticus* PI13, *L. gasseri* LA806, *Lc. lactis* PI23, *B. bifidum* PI22, and *B. longum* PI10 (*p* < 0.05 to 0.0001). The probiotic-induced increase in GLP-1 production was higher or similar to the induction obtained after incubation of endocrine cells with butyrate or acetate. Two SCFAs reported to enhance GLP-1 secretion [36]. *B. longum* PI10 and *B. bifidum* PI22 were the most potent GLP-1 inducers (*p* < 0.0001).

We have previously screened the same probiotic strains for their anti-inflammatory activity and their ability to restore the intestinal epithelial barrier [20]. Figure 3 combines the results obtained with all these in vitro models with the present ones (i.e., ability to limit lipid accumulation in adipocytes and to enhance GLP-1 production by endocrine cells). Based on these screenings, we selected the following nine strains: one strain with only the most anti-inflammatory property (*La. rhamnosus* LA305), one strain with high ability to strengthen the gut barrier (*L. helveticus* PI5), and one strain enhancing GLP-1 only (*L. helveticus* PI13). In addition, strains combining two properties (*B. breve* LA805, *Li. salivarius* LA307, *L. gasseri* LA806), three properties (*B. animalis* subsp. *lactis* LA804, *B. bifidum* PI22), or all properties (*B. longum* PI10) were also selected for further in vivo testing.

### 3.3. Selected Strains Limit High-Fat Diet-Induced Body Weight Gain and Fat Accumulation

Using a murine model of high-fat diet (HFD)-induced obesity, we first evaluated the impact of the administration of the individual strains that were selected based on their high abilities *in vitro*. None of the selected strains administered alone was able to limit the development of obesity (data not shown) with the exception of the *B. longum* subsp. *longum* (*B. longum*) PI10 strain, which is a highly potent inducer of GLP-1 and IL-10 that also significantly decreased the accumulation of lipids in adipocytes and restored the gut barrier.

We, therefore, conducted another experiment to confirm the effect of the *B. longum* PI10 strain alone. We also evaluated the impact of different mixtures including *B. animalis* subsp. *lactis* LA804, the best inducer of IL-10 that like *B. longum* PI10, also exhibited a good ability to restore the intestinal barrier and to induce the release of GLP-1. *B. animalis* subsp. *lactis* LA804 was combined with the following lactobacilli: *L. helveticus* PI5, which best strengthened the epithelial barrier, *L. gasseri* LA806, which showed a good ability to restore the gut barrier and to induce GLP-1, or *L. helveticus* PI13, a good inducer of GLP-1 (Figure 4).

As expected, HFD-fed mice gained significantly (*p* < 0.0001) more weight than LFD-fed mice during the dieting period (Figure 4A). At sacrifice (14 weeks post-diet), HFD-fed mice weighed more than the LFD-fed counterparts (*p* < 0.01) (Figure 4B) and showed a significant (*p* < 0.01) increase in epididymal and subcutaneous white adipose tissue masses (respectively, EWAT and SCAT, Figure 4C,D).

The administration of *B. longum* PI10 alone and the combination of *B. animalis* subsp. *lactis* LA804 and *L. gasseri* LA806 significantly limited HFD-induced body weight gain (*p* < 0.05) and HFD-induced increased EWAT mass (*p* < 0.05), while it had no significant impact on SCAT mass (Figure 4A–D). In contrast, the other mixtures did not exhibit protective effects under these conditions. It has to be noted that none of the probiotic strains, either administrated alone or in combination, has an effect on food intake (Figure 4E).

Expectedly, fasting blood glucose and leptin levels were significantly higher in HFD-fed mice than in LFD-fed mice (respectively, *p* < 0.01 and *p* < 0.05) (Figure 4F,G). Consistent with their impact on body weight and fat mass, treatment with *B. longum* PI10 alone or with the combination of *B. animalis* subsp. *lactis* LA804 and *L. gasseri* LA806 significantly limited the HFD-induced increase in blood glucose and leptin levels. Despite no significant impact on body weight or fat mass (Figure 4A–D), the administration of the mixture of *B. animalis* subsp. *lactis* LA804 and *L. helveticus* PI5 also significantly decreased blood leptin levels (*p* < 0.05). Blood insulin and adiponectin levels were not significantly different between the different groups (data not shown). To evaluate the impact of the probiotics on the development of insulin resistance, an intra-peritoneal glucose tolerance test (IP-GTT) was performed after 12 weeks of diet (Figure 4H). Compared to LFD-fed animals, HFD-fed mice were strongly glucose intolerant (*p* < 0.001). However, none of the probiotic strains or mixtures had an effect on the in vivo tolerance to glucose.

### 3.4. Selected Strains Alleviate High-Fat Diet-Induced Fat and Gut Inflammation

To decipher the mechanisms of action of the selected strains on obesity, we characterized the EWAT’s inflammatory status (Figure 5A). As expected and previously reported [12], HFD-fed mice showed a strong and significant increase in the expression of inflammation-related genes (*tnfa*, *mcp1* and *cd68* (*p* < 0.01). Importantly, the administration of *B. longum* PI10 alone or the combination of *B. animalis* subsp. *lactis* LA804 and *L. gasseri* LA806 (Mixture) significantly impaired the HFD-induced increase of inflammatory gene expression (*p* < 0.05 or 0.01).

The anti-inflammatory effect of the selected strains was also observed at the intestinal level. Mice treated with *B. longum* PI10 alone or the mixture of *B. animalis* subsp. *lactis* LA804 and *L. gasseri* LA806 showed a significant decrease in the *Mcp1* gene expression in the jejunum, as compared to untreated HFD-fed animals (Figure 5B). As previously reported [12], jejunal expression of the gene encoding FABP1, which facilitates fatty acid (FA) binding, was significantly decreased upon probiotic treatment. The expression of the scavenger receptor CD36, involved in FA transport by enterocytes was decreased only in the mice that received the mixture compared to obese, untreated mice in which the expression of these genes was upregulated in comparison to lean mice, yet not significantly. The mixture also induced a significant increase in the duodenal mRNA expression of the gene encoding the Takeda G-protein coupled receptor 5 (*tgr5*), a receptor for bile acids, as well as of that of proglucagon (*gcg*) (Figure 5C). However, we did not observe any changes in the portal blood levels of the gut peptides GLP-1 and the gut hormone peptide YY (data not shown).

### 3.5. Selected Strains Impact on the Hypothalamic Regulation of Energy Balance

We observed a strong and significant increased expression of the genes encoding leptin (*lep*) and leptin receptor (*lepr*) in the hypothalamus of mice having received either *B. longum* PI10 alone or the mixture (Figure 6A,B). This was associated with (i) a significant increase in the expression of the gene encoding the hypocretin neuropeptide precursor (*hcrt*) also called orexin (Figure 6C), which is known to increase insulin sensitivity and limit body weight gain [37] and (ii) a moderate increase in the expression of the gene encoding the anorexigenic proopiomelanocortin (*pomc*) (Figure 6D) [38]. However, the expression of all these genes was not significantly decreased in obese mice compared to lean mice. No significant changes in the expression of the genes encoding the orexigenic agouti related protein (*agrp*) and neuropeptide Y (*npy*) were observed (data not shown).

### 3.6. The Selected Strains Impact on the Composition of the Gut Microbiota

As expected from literature, the gut microbiota is dominated by four main phyla: Firmicutes, Bacteroidetes, Proteobacteria, and Actinobacteria (Figure 7A). The ratio Bacteroidetes/Firmicutes (Figure 7A) and the abundance of Actinobacteria (Figure 7B, *p* < 0.05) were clearly decreased in HFD-fed mice, as previously reported [12], while the proportion of proteobacteria increased. In mice treated with *B. longum* IP10 alone and the mixture of *B. animalis* subsp. *lactis* LA804 and *L. gasseri* LA806, the level of Actinobacteria was significantly (*p* < 0.05 and 0.001, respectively) restored and that of Proteobacteria decreased. The ratio Bacteroidetes/Firmicutes was partially restored in mice that received the mixture (Figure 7A,B). PCoA analysis revealed that bacterial communities varied between groups of mice (Figure 7B). However, the α diversity was not significantly different between the groups (observed and Simpson) (Figure 7D). At the family level (Figure 7E,F, Appendix A), obese mice were characterized by a significant decrease in Bifidobacteriaceae (*p* < 0.01) and Alcaligenaceae (*p* < 0.05) and an increase in Clostridiaceae, Peptostreptococcaceae, and Streptoccocaceae (*p* < 0.05). At the genus level (Figure 7G, Appendix A), there was a significant decrease in *Eubacterium rectale*, *Bifidobacterium*, *Eisenbergiella*, *Faecalibacterium*, *Rikenellaceae*, *Ruminococcaceae* (*p* < 0.01), and *Lachnospiraceae* (*p* < 0.05). In contrast, there was a significant increase in the abundance of *Blautia, Lactococcus, Peptoclostridium, Peptococcus, Ruminiclostridium, Ruminococcus, Turicibacter* (*p* < 0.05), *Oscillobacter*, and *Parabacteroides* (*p* < 0.01). In mice treated with *B. longum* IP10, the abundance of *Bifidobacteriaceae*/*Bifidobacterium* and *Ruminococcaceae*/*Ruminococcus* was significantly restored (*p* < 0.05–0.01) in comparison with untreated obese mice. On the contrary, the levels of *Blautia*, *Lachnoclostridium*, *Parabacteroides* (*p* < 0.05), and *Roseburia* (*p* < 0.001) were significantly decreased. The protective effect of the mixture of *B. animalis* subsp. *lactis* LA804 and *L. gasseri* LA806 was associated with a significant increase in Actinobacteriaceae/*Bifidobacteriaceae*/*Bifidobacterium* (*p* < 0.01–0.001) and *Coriobacteriaceae* (*p* < 0.01) and a drop of *Clostridium* sensu stricto 1# and *Oscillobacter* (*p* < 0.01).

### 3.7. The Selected Strains Modify SCFA Production and Bile Acid Metabolism

Since changes in the composition of the gut microbiota may alter its metabolic activity, we next measured in cecal contents the levels of total SCFAs and of the main ones (acetate, propionate, and butyrate) known to exert multiple beneficial effects on mammalian energy metabolism. As expected, total SCFAs and acetate levels were significantly lower in HFD-fed mice than in LFD-fed animals, whereas the reduced levels of propionate and butyrate were not significant (Figure 8). The mixture of *B. animalis* subsp. *lactis* LA804 and *L. gasseri* LA806 significantly restored the levels of total SCFAs and acetate, and, to a lesser extent, the levels of propionate and butyrate. In contrast, *B. longum* PI10 had no impact on SCFAs’ levels.

To further explore the mechanisms through which the selected strains improved the metabolic phenotype, we assessed their ability to modify the bile acid (BA) pool, both in the portal blood (Figure 9, Appendix A) and cecal content (Figure 10, Appendix A). In the portal blood, HFD significantly decreased total BA concentrations (Figure 9A) via a significant decrease in Total ursodeoxycholic acid (UDCA) (which is due to decreased free UDCA but not tauro-conjugated TUDCA, Appendix A) and a tendency, albeit not significant, via a decrease of Total cholic acid (CA) and all the muricholic acid (MCA) species (Total αMCA, Total βMCA, and Total ωMCA) (Figure 9D). This is notably due to a decrease in the free forms of αMCA and βMCA (Appendix A). However, the ratio of free/conjugated BAs, reflecting deconjugations of BAs by the microbiota, was not modified (Figure 9B) between HFD-fed and LFD-fed mice nor of the ratio primary/secondary BAs, reflecting transformation of BAs by the microbiota (Figure 9C). HFD increased the ratio 12αOH/no12αOH BA (data not shown), which is a marker of insulin-resistance. The probiotics did not change total BA concentrations, nor the ratios. Thus, it seems that the selected strains do not alleviate the HFD-induced BA changes in the portal vein.

In the cecal content, BAs are presented as percentages of the total BA pool (Figure 10, Appendix A). HFD had no effect on the free/conjugated (Figure 10A) nor primary/secondary (Figure 10B) BA ratios. However, HFD significantly increased the percentage of Total DCA (Figure 10C) via an increase in its free form (Appendix A) while simultaneously decreasing the percentage of free βMCA (Appendix A) and, thus, Total βMCA (Figure 10C). Whereas probiotic strains had no effects on BAs in the portal blood, they affected BA proportions in the cecal content (Figure 10C, Appendix A). Both *B. longum* PI10 and the *B. animalis* subsp. *lactis* LA804 and *L. gasseri* LA806 mixture decreased the free ωMCA (Appendix A) and, consequently, the Total ωMCA (Figure 10C). In addition, the *B. animalis* subsp. *lactis* LA804 and *L. gasseri* LA806 mixture increased the free βMCA (Appendix A) and, consequently, the Total βMCA (Figure 10C), counteracting the HDF-induced decrease of this BA species (Figure 10C, Appendix A).

## 4. Discussion

The persistence of an altered microbiome, following cycles of obesity and dieting, has been shown to contribute to enhanced metabolic deregulation following body weight regain [10], suggesting a strong rationale for targeting microbiota dysbiosis in the management of obesity. We and others have reported that potential probiotic strains can limit body weight gain in different rodent models of obesity [12,13,14], while other studies have observed that probiotics have little or no effect and, in some cases, can even worsen body weight gain [39,40]. Even if selected probiotics were reported to have beneficial effects in obese individuals, affecting body mass index and fat mass (for review, see Reference [41]), the investigation of probiotics in clinical trials targeting obesity and diabetes remains rare, with limited sample sizes and no follow-up studies to clearly demonstrate the long-term impact of such treatment. Strains that have shown beneficial effects in obesity and T2DM in humans mainly belong to the former genus *Lactobacillus* [42,43]. However, it has been reported that lactobacilli are more abundant in obese and diabetic patients, even though this increase could be an indirect compensatory response [16]. Moreover, anti-obesity effects of probiotics are certainly strain and/or-dose-dependent, and mechanisms involved are still to be identified. It, thus, remains important to better understand strain-specificity of probiotics and design new tools and criteria for the selection of effective strains.

In this study, we targeted the known microbiota and metabolic dysfunctions associated with obesity and screened a large collection of potential probiotic strains, including 15 lactobacilli and 6 bifidobacteria of different species for their beneficial effects. Obesity is mainly characterized by an increase in body fat mass and lipid accumulation, associated with chronic low-grade inflammation that triggers insulin resistance [6]. Modification of the gut microbiota has an impact on gut permeability that leads to metabolic endotoxemia and, therefore, contributes to obesity-associated inflammation, leading, in turn, to the development of insulin resistance [39]. We have previously evaluated in vitro the anti-inflammatory abilities of selected strains and their ability to improve gut barrier integrity [20]. Bifidobacteria were among the strains with the best anti-inflammatory profile (with the exception of *B. animalis* subsp. *lactis* LA306) while three lactobacilli had moderate but significant anti-inflammatory properties (*L. salivarius* LA307, *L. casei* PI20 and *La. rhamnosus* LA305) [20]. Four probiotics (*L. helveticus* PI5, *L. acidophilus* PI11, *B. animalis* subsp. *lactis* LA804, and *L. gasseri* LA806) not only restored but also strengthened the barrier [20]. In the present study, we extended their *in vitro* functional effects by evaluating their ability to limit lipid accumulation in mature adipocytes and to favor the secretion of gut peptides by enteroendocrine cells. The intracellular accumulation of lipids in 3T3-L1 adipocytes decreased upon stimulation with 14 out of 23 probiotic strains, but only four strains (*Li. salivarius* LA307, *B. longum* PI10, *Li. salivarius* PI2, and *Lc. lactis* PI23) had a statistically significant impact, indicating that these strains might favor lipolysis and have anti-adipogenic activities. Even if lipoprotein lipase (LPL) is known to promote adipogenesis in the early stage of adipose differentiation, it is a critical determinant of plasma triglyceride (TG) clearance. While LPL is widely distributed in a variety of tissues, it is translocated to capillary endothelial cells in muscle and adipocytes and could attach to their luminal surface to hydrolyze TG [40,41]. Therefore, increasing lipolysis in adipocytes is now considered a useful therapeutic target for treating obesity [42]. PPARγ is an important transcription factor in the development and function of the adipose tissue. Even if it is involved in adipocyte differentiation and increased storage of fatty acids during adipogenesis, it plays a central role in maintaining insulin sensitivity, notably by activating the expression of adiponectin, involved in the sensitization of liver and muscle to insulin. It can also antagonize the function of pro-inflammatory transcription factors, such as nuclear factor-κB (NF-κB), thereby, leading to a decrease in pro-inflammatory cytokines and inflammatory status of the adipose tissue [43].

The end-products of the fermentation of fibers, in particular SCFAs, act as signaling molecules capable of inducing the release of intestinal peptides by enteroendocrine cells, such as GLP-1 and PYY. These peptides are known to be involved in satiety, in the control of gut barrier function, in glucose and energy homeostasis, and, consequently, insulin sensitivity [44,45]. Thirteen strains induced a significant secretion of GLP-1 upon stimulation of STC-1 cells, *B. bifidum* PI22, and *B. longum* PI10 being the most powerful inducers.

The combination of these in vitro screenings allowed the selection of nine strains that had the best efficacy in one or several tests for further evaluation of their ability to limit the development of obesity in a murine model of HFD-induced obesity. None of the strains had a protective effect when administered as a single strain, with the exception of *B. longum* PI10. This strain was the best inducer of GLP-1 and one of the best inducers of the anti-inflammatory cytokine IL-10. Strain PI10 was also able to significantly decrease lipid accumulation in adipocytes and had a moderate impact on epithelial permeability. Since combining bifidobacteria and lactobacilli could also represent an interesting strategy [46], we evaluated the impact of three mixtures. *B. animalis* subsp. *lactis* LA804, which was the best inducer of IL-10 and had a good capacity to restore the epithelial barrier and to induce the release of GLP-1, was combined with different lactobacilli that either exhibited good ability to strengthen the epithelial barrier (*L. helveticus* PI5, *L. gasseri* LA 806) or to induce the release of GLP-1 (*L. helveticus* PI13, *L. gasseri* LA806).

We confirmed the positive effect of *B. longum* PI10 alone and highlighted the beneficial effects of the combination of *B. animalis* subsp. *lactis* LA804 and *L. gasseri* LA806 in limiting body weight gain and adiposity, especially the epididymal adipose tissue mass. While these treatments ameliorated the inflammatory status of mice and improved fasting blood glucose and leptin levels, we did not observe any improvement of the HFD-induced glucose intolerant state. A significant decrease in inflammatory markers, in particular those corresponding to macrophages, was observed in visceral adipose tissues and in the jejunum of bacteria-treated mice. The beneficial effects of these treatments were also associated with a significant decrease in the jejunal gene expression of the fatty acid binding protein (FABP)1, which is involved in the binding, transport, and metabolism of long-chain FA, and which was up-regulated in obese mice. As lipid chaperones, FABPs facilitate the transport of lipids to cellular specific compartments, such as to the formation of lipid droplets for storage. FABPs are closely associated with both metabolic and inflammatory processes and it is noteworthy that mice with *FABP*-deficiency in the gut have been reported to gain more weight. Therapeutically targeting FABP has been suggested as a novel strategy in the management of metabolic diseases [47].

Protection was also correlated to an increase in the expression of genes encoding proglucagon and bile acid receptor TGR5 in the duodenum. The gastro-intestinal tract is now considered as the largest endocrine organ of the body, able to produce various hormones and neuropeptides involved in food intake, energy expenditure, and glucose homeostasis. In particular, the classical role of proglucagon, produced partly by the gut endocrine L-cells, is to maintain glucose homeostasis, but also to regulate food intake and energy metabolism. Proglucagon is the precursor of both GLP-1 and GLP-2. Alike incretin, GLP-1 facilitates insulin secretion in a glucose-dependent-manner and is also involved in food intake control [6]. It is, therefore, becoming an important target for the treatment of obesity. The abundance of these gut peptides has been shown to increase upon prebiotic treatment and be related to enhanced glucagon expression [48]. It is important to note that the strains that exhibited the best protective effect (*B. longum* PI10) was also the strongest in vitro inducer of GLP-1, as shown using the murine STC-1 endocrine cell line. *L. gasseri* LA806 that was included in the protective mixture was also a good inducer of GLP-1. Unfortunately, although selected strains increased the expression of proglucagon in vivo, we were unable, under our experimental conditions, to detect changes in the portal blood level of GLP-1. Since the action of GLP-1 is glucose-dependent, we could speculate that we were not in the best conditions to measure its level, since we did not administer glucose before the mice were sacrificed. Moreover, active GLP-1 has a very short half-life due to its rapid degradation in the blood by dipeptidyl peptidase-4 (DPP-4), and is found in low concentrations [49]. Although DPP-4 inhibitors have been added to the samples, it is possible that GLP-1 concentrations were under evaluated. Lactobacilli and bifidobacteria exert proteolytic activity thanks to their cell envelope-associated peptidases (CEPs) and other peptidases. The proteolytic activity depends on CEPs and peptidase profiles, which vary from one strain to another [50]. Preliminary data indicated that most of the strains we tested had a DPP4-like activity that could be responsible for the degradation of GLP-1 (see Appendix A), at least in our *in vitro* model, since, in vivo, we can expect that the basal secretion of GLP-1 could not be degraded by DPP-4 release in the gut lumen.

Bile acids have emerged as key signaling molecules, notably through their interaction with specific receptors, such as the Farnesoid X Receptor (FXR) and TGR5. TGR5 signaling is also involved in the release of GLP-1 and, as such, in insulin sensibility [51]. TGR5 is now considered as a promising target for the development of a potential therapeutic intervention in metabolic diseases, i.e., obesity and T2DM [52]. The gut microbiota has a unique influence on the diversity of the bile acid pool, notably through deconjugation and de-hydroxylation or epimerization, with significant consequences for the host that can either favor health or disease [53]. Bile salt hydrolase (BSH) is one of the main microbial enzymes that contribute to bile acid metabolism by generating secondary bile acids [54]. BSH activity is specific to the microbiota and has been mainly characterized in lactobacilli [55] but also in bifidobacteria [56]. We, thus, can speculate that the strains we studied could have BSH activity, and, consequently, have an impact on the composition of bile acid pool and, therefore, on TGR5 activity and GLP-1 production. Since TGR5 agonists may induce unwanted side effects, such as cardiovascular effects, gallstone formation and gallbladder emptying effects [57], changes in the composition of bile acid pool through manipulation of the gut microbiota could represent an interesting alternative. This observation may be an additional element important in the ongoing discussion of the controversial impact of bacterial BSH in host’s health [58]. In the cecal content, *B. longum* PI10 increased the proportion of Total βMCA via increases of its free form (βMCA), but not its tauro-conjugated form (TβMCA), an antagonist of the Farnesoid-X-receptor (FXR) [59], which inhibits GLP-1 synthesis and secretion in the intestine [60]. Moreover, the decrease of Total ωMCA observed with *B. longum* PI10 and the combination of *B. animalis* subsp. *lactis* LA804 and *L. gasseri* LA806 are not related to FXR or TGR5 activity alterations, as this BA species is not a ligand of these receptors. Thus, we can hypothesize that the improvement of metabolic homeostasis with these probiotics is not due to BA changes.

As expected from the literature [2], we have observed a dysbiotic microbiota in the intestine of mice fed a high-fat diet in comparison to mice fed with a low fat diet. The dysbiosis was characterized by an increased relative abundance of *Firmicutes* compared to *Bacteroidetes*, an increased proportion of Proteobacteria, and a strong drop of Actinobacteria. Numerous studies support the concept that increased prevalence of *Proteobacteria* is linked to an unstable gut microbial community [61], while Bifidobacteria (belonging to the Actinobacteria) can be considered as a marker of gut health and have been shown to improve diabetes induced by high-fat-diets in mice through a mechanism associated with endotoxemia [62]. Supplementation with *B. longum* PI10 alone or *B. animalis* subsp. *lactis* LA804 and *L. gasseri* LA806 mixture changed the bacterial communities. The two supplementations significantly restored the Actinobacteria/Bifidobacteriaceae ratio, decreased the proportion of Proteobacteria, and modified the microbiota composition at the family and genus levels.

Recent advances have reported the interaction between the microbiota and gut-brain axis, influencing neural, endocrine, and immune functions [63]. Leptin plays a crucial role in the regulation of energy, glucose, and lipid homeostasis, as well as neuroendocrine- and immune functions. Leptin levels are positively correlated with the amount of body fat, and, as such, obese individuals have higher levels of leptin in adipose tissue and elevated circulating leptin levels. In line with this, we observed an increase in leptin levels in HFD-fed mice together with an increase in fasting blood glucose levels, which were both significantly reduced upon probiotic administrations. Leptin exerts its anorexic effects after its binding to specific receptors located in the central nervous system, mostly in the hypothalamus. However, the elevated blood leptin levels observed in obese individuals fail to reduce excess adiposity, suggesting leptin resistance [64]. Disruption of leptin signaling in the neuronal circuits of the brain can include impaired leptin transport across the blood-brain barrier and hypothalamic inflammation [65]. The protective effects of the strains were associated with an increase in gene expression of leptin, leptin receptor, and hypocretin in the hypothalamus. Leptin, by inhibiting the hypothalamic-pituitary-adrenal axis, can act as a modulator of the neuronal activity of hypocretin that plays a role in satiety and energy intake [66]. Proopiomelanocortin neurons in the hypothalamus are also targets for leptin and the expression of the *pomc* gene in the central nervous system induces the production of melanocortin peptides, which appear to play a significant role in appetite and the control of body weight. The gene expression of *pomc* was also reduced in probiotic-treated mice, even if this was not statistically significant.

## 5. Conclusions

Overall, the multi-criteria *in vitro* screening strategy we used was successful to identify at least one high-potential strain, *B. longum* PI10, able to significantly impact obesity as a single strain, as well as a high-potential mixture composed of a strain of *B. animalis* subsp. *lactis* and a strain of *L. gasseri*. These potential probiotic treatments may act through multiple mechanisms, including a decrease in the inflammatory status and improvement of fasting glycemia and leptin resistance. We also identified other potential mechanisms involving bile acid pool modification and TGR5 activation. The impact on GLP-1 induction could be one of the central mechanisms, since the *B. longum* PI10 and *L. gasseri* strain included in the mixture were the most potent inducers of GLP-1 *in vitro*. It will be important to clarify this mechanism in vivo in the future by determining the pharmacokinetics of identified mediators. Finally, the protective impact of the two probiotics was associated with changes of the gut microbiota composition, SCFAs, and maybe bile acid profiles. These results provide crucial tools for selection of the best strains or mixtures for the development of more effective approaches in the management of obesity and metabolic diseases. They also bring substantial insight into how the host-microbial interaction may govern their protective effects.

## Figures and Tables

**Figure 1 nutrients-13-00713-f001:**
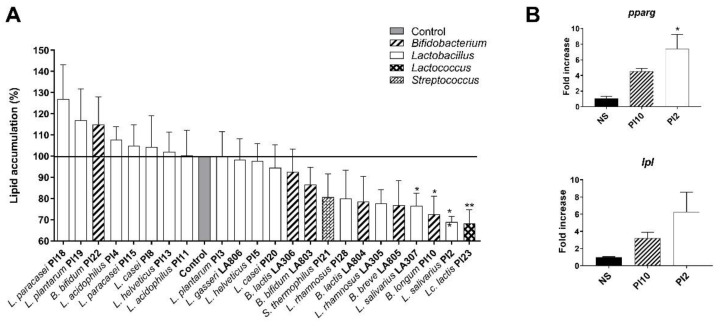
Effect of selected strains on adipocyte lipid accumulation. (**A**). Intracellular lipids were quantified after Oil-red-O staining. The dye was eluted from cells with isopropanol and quantified. Results are expressed as a percentage from control adipocytes. * refer to the comparison of probiotic-stimulated 3T3-L1 versus untreated cells (medium control). (**B**). Relative mRNA expression (RT-qPCR) of *pparg* and *lpl* in *B. longum* PI10- and *L. salivarius* PI2-treated adipocytes. * refer to the comparison of probiotic-stimulated 3T3-L1 versus untreated cells (medium control), * *p* < 0.05, ** *p* < 0.01.

**Figure 2 nutrients-13-00713-f002:**
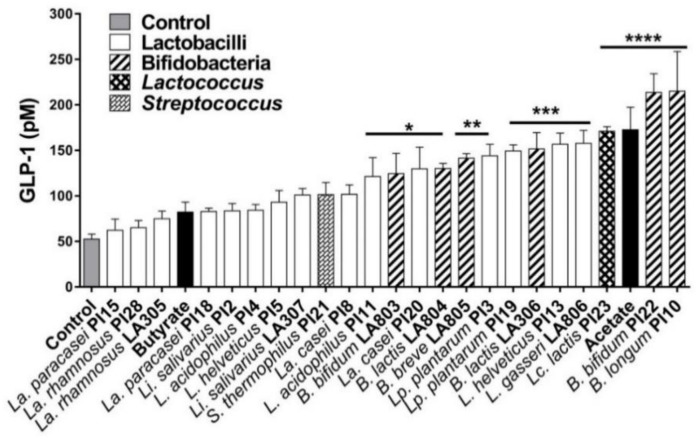
Effect of selected strains on GLP-1 secretion by STC-1 cells. Cells were stimulated for 8 h with probiotics. GLP-1 secretion was measured by Radioimmunoassay (RIA). * refers to the comparison between bacteria-exposed STC-1 cells and untreated cells (medium control), * *p* < 0.05, ** *p* < 0.01, *** *p* < 0.001, **** *p* < 0.0001.

**Figure 3 nutrients-13-00713-f003:**
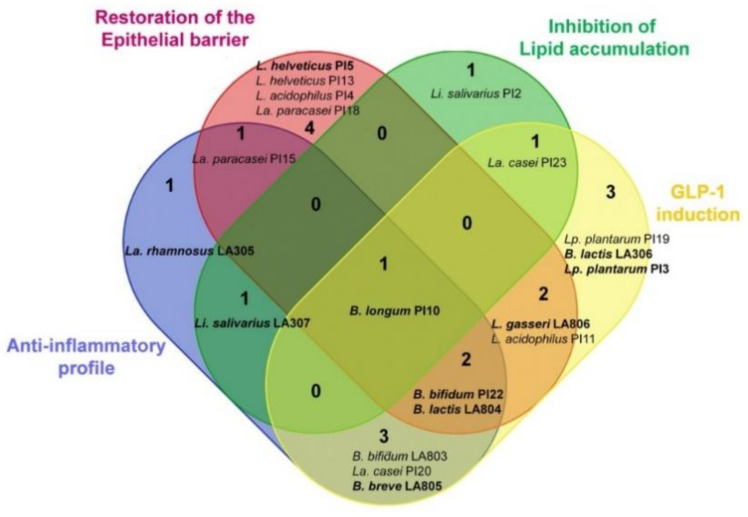
Venn diagram summarizing the in vitro functional properties of the most effective strains and highlighting the strains that combine different properties. The strains with the highest measured abilities in one or several models are shown in bold.

**Figure 4 nutrients-13-00713-f004:**
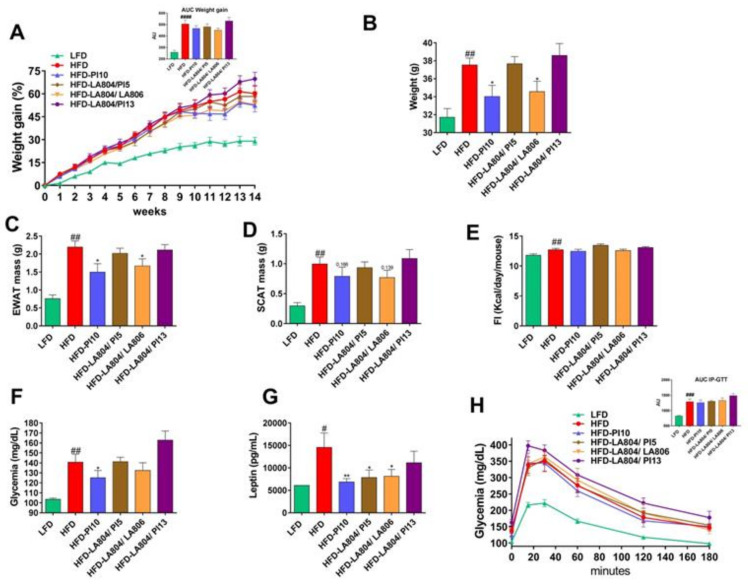
*B. longum* PI10 and the *B. animalis* subsp. *lactis* LA804/*L. gasseri* LA806 mixture limited high-fat diet (HFD)-induced body weight gain, fat mass increase, hyperglycemia, and hyperleptinemia but had no impact on HFD-induced glucose intolerance. (**A**) Evolution of body weight gain (in %) in mice receiving probiotics or excipient and fed a low-fat diet (LFD) or HFD for 14 weeks and the corresponding area under the curve (AUC) (in arbitrary unit [AU]). (**B**) Body weight at sacrifice (in g). (**C**) Epididymal adipose tissue (EWAT) mass at sacrifice (in g). (**D**) Subcutaneous adipose tissue (SCAT) mass at sacrifice (in g). (**E**) Cumulative food intake (in kcal/day/mouse). (**F**) Fasting blood glucose levels at 13 weeks of diet (in mg/dL). (**G**) Fasting blood leptin level at 13 weeks of diet (in pg/mL). (**H**) Intra-peritoneal glucose tolerance test (IP-GTT) at 12 weeks of diet and the corresponding AUC (in AU). Blood glucose levels (mg/dL) measured after intraperitoneal glucose injection. Data are expressed as mean ± S.E.M. of 4 (LFD) to 8 (HFD) mice per group. ^#^ corresponds to diet effect (HFD vs. LFD), * corresponds to treatment effect (Probiotic vs. Excipient). * or ^#^
*p* < 0.05, ** or ^##^
*p* < 0.01; ^###^
*p* < 0.001; ^####^
*p* < 0.0001.

**Figure 5 nutrients-13-00713-f005:**
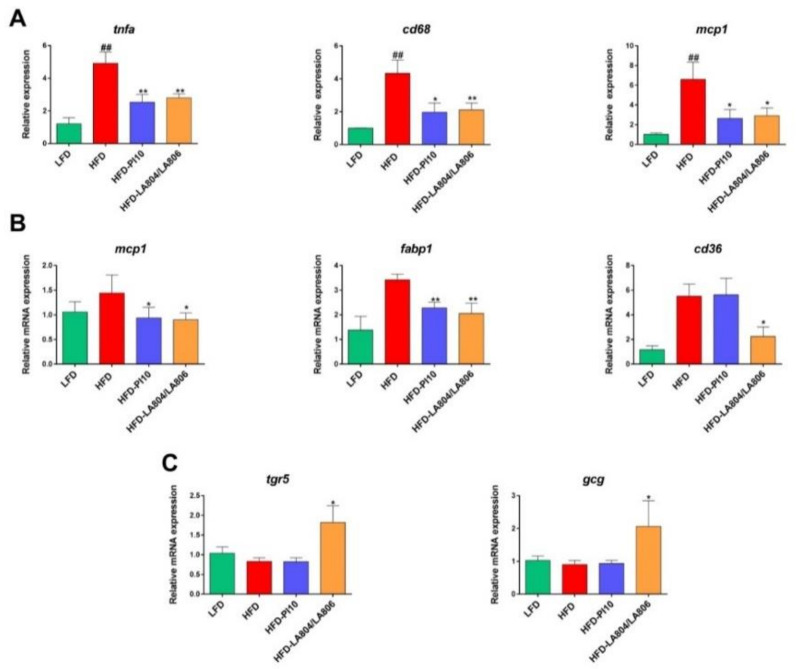
*B. longum* PI10 and the mixture of *B. animalis* subsp. *lactis* LA804 and *L. gasseri* LA806 limited the HFD-induced gene expression of inflammatory markers in the visceral adipose tissue and modulated the HFD-induced expression of genes involved in inflammation, fatty acid metabolism, and bile acid signaling in the small intestine. (**A**) The expression levels of inflammatory-related (*tnfa, cd68* and *mcp1)* were quantified in the visceral adipose tissue at sacrifice by RT-qPCR. (**B**) Quantification of the expression levels of genes involved in inflammation (*mcp*-*1*) and fatty acid uptake and transport (*fabp1*, *cd36*) in the jejunum. (**C**) The expression levels of genes encoding the receptor for bile acids (*tgr5*) and proglucagon (*gcg*) in the duodenum. Values are expressed as the relative mRNA levels compared with LFD mice and expressed as mean ± standard error of the mean (S.E.M) of 6 (LFD group) to 8 (HFD groups) mice per group. #corresponds to regime effect (HFD vs. LFD), * corresponds to treatment effect (Probiotic vs. Excipient), * *p* < 0.05, ** or ^##^
*p* < 0.01.

**Figure 6 nutrients-13-00713-f006:**
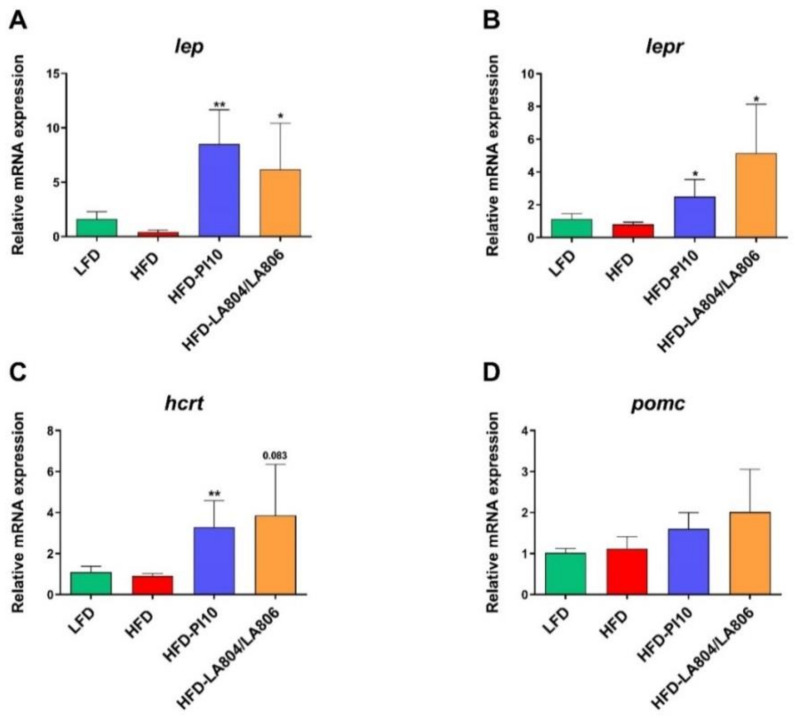
*B. longum* PI10 and the mixture of *B. animalis* subsp. *lactis* LA804 and *L. gasseri* LA806 modulated hypothalamic expression of genes involved in food intake and energy expenditure. The expression levels of genes encoding (**A**), leptin (*lep*), (**B**) leptin receptor (*lepr*) (**C**) orexin (*hcrt*), and (**D**) proopiomelanocortin (*Pomc*) were quantified in the hypothalamus of LFD-fed mice and HFD-fed mice treated or not with the selected probiotic strains by RT-qPCR. Values are expressed as the relative mRNA levels compared with LFD mice and expressed as mean ± SEM of 6 (LFD group) to 8 (HFD groups) mice per group. *corresponds to treatment effect (Probiotic vs. Excipient), * *p* < 0.05, ** *p* < 0.01.

**Figure 7 nutrients-13-00713-f007:**
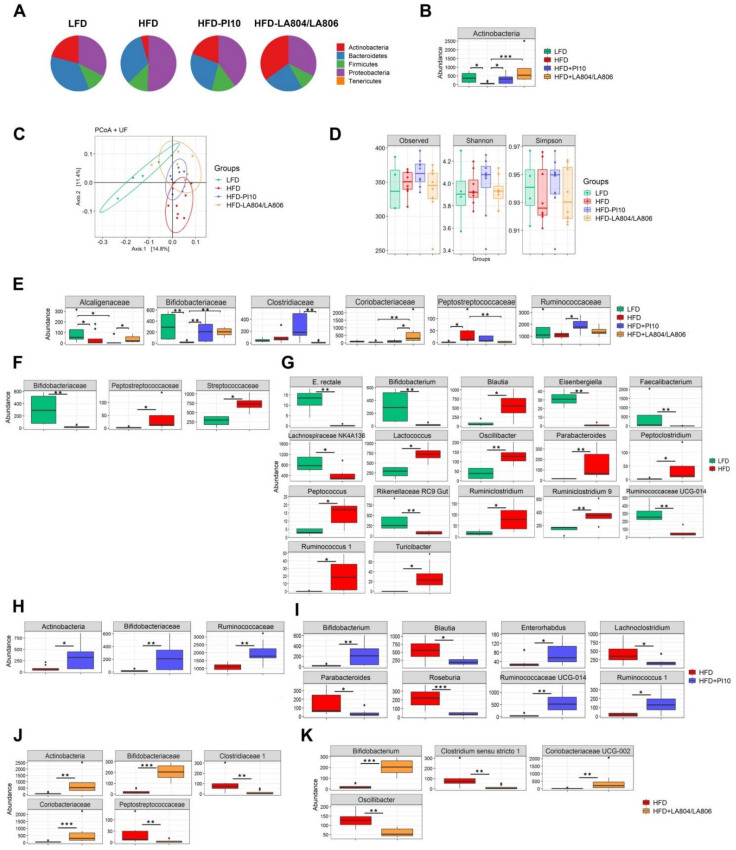
*B. longum* PI10 strain and *B. animalis* subsp. *lactis* LA804 and *L. gasseri* LA806 mixture had an impact on the HFD-induced microbiota dysbiosis. (**A**) Relative abundance of the dominant phyla evaluated by 16S rRNA MiSeq sequencing in cecal contents. (**B**) Relative abundance of Actinobacteria for each group of mice. (**C**) PCoA (Unifrac) analysis showing comparative differences between groups at the phylum, family, and genus levels. (**D**) Diversity indexes (observed, Shannon, Simpson). (**E**) Differences in the relative abundance of families between groups. (**F**) Differences in the relative abundance of families and (**G**) genera between LFD and HFD groups. (**H**) Differences in the relative abundance of families and (**I**) genera between HFD and HFD-PI10 groups. (**J**) Differences in the relative abundance of families and (**K**) genera between HFD and HFD-LA804 + LA806 groups. * *p* < 0.05, ** *p* < 0.01; *** *p* < 0.001.

**Figure 8 nutrients-13-00713-f008:**
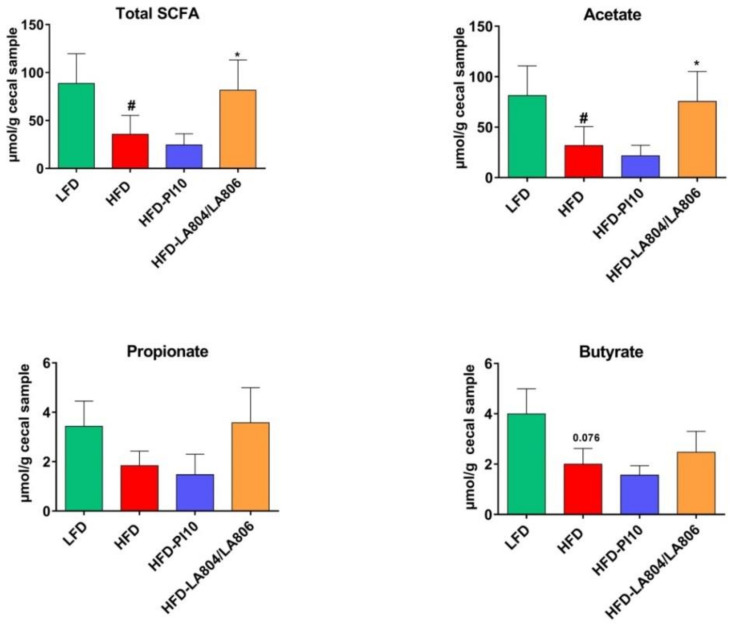
The mixture of *B. animalis* subsp. *lactis* LA804 and *L. gasseri* LA806 significantly restored the production of SCFAs in the cecum. Levels of total SCFAs, acetate, propionate, and butyrate are expressed in µmol/g of cecal content as mean ± SEM of 6 (LFD group) to 8 (HFD groups) mice per group. ^#^ corresponds to the diet effect (HFD vs. LFD), * corresponds to the treatment effect (Probiotic vs. Excipient), * or ^#^
*p* < 0.05.

**Figure 9 nutrients-13-00713-f009:**
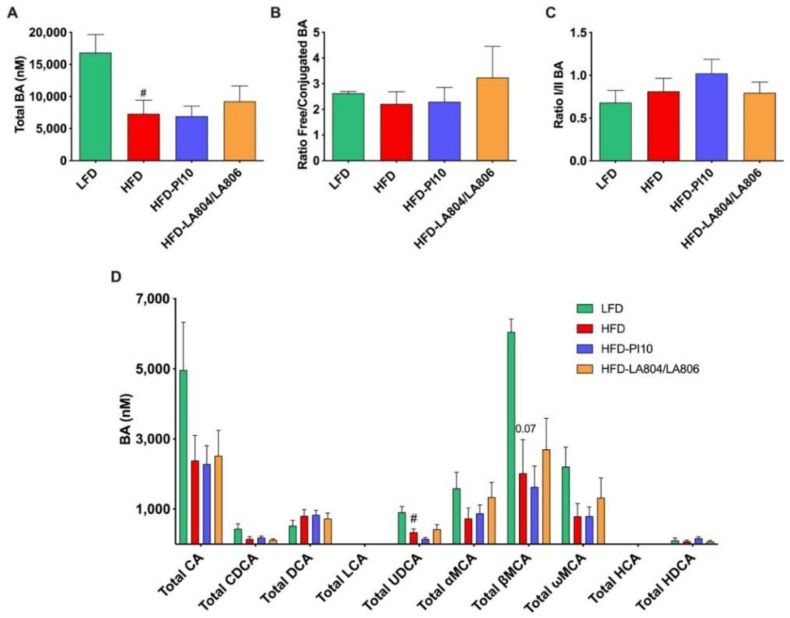
*B. longum* PI10 strain and *B. animalis* subsp. *lactis* LA804 and *L. gasseri* LA806 mixture have no effect on portal BA concentrations. (**A**) Total BA concentrations; (**B**) Ratio Free/Conjugated BA; (**C**) Ratio primary/secondary BA; (**D**) Different BA species concentrations. Results are expressed as mean ± SEM. ^#^
*p* < 0.05 HFD vs. LFD (Mann-Whitney test).

**Figure 10 nutrients-13-00713-f010:**
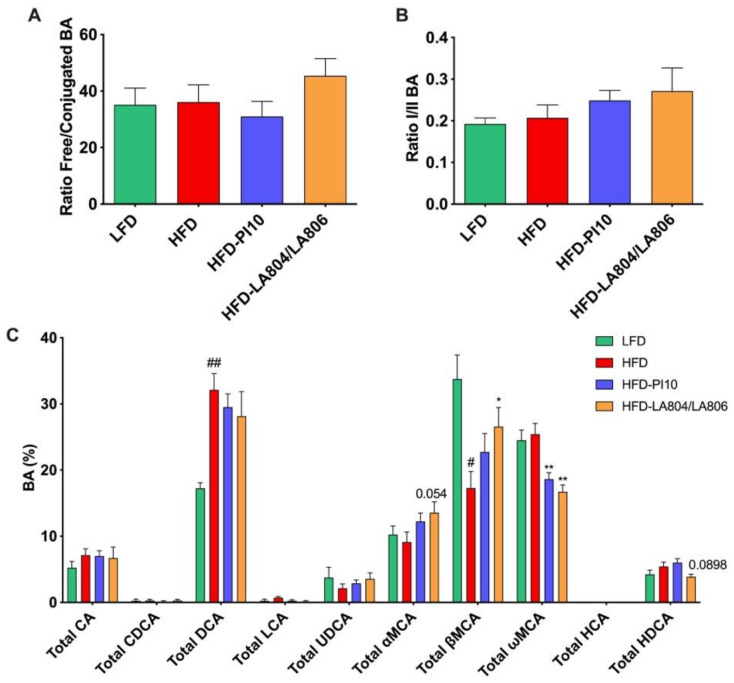
*B. longum* PI10 strain and *B. animalis* subsp. *lactis* LA804 and *L. gasseri* LA806 mixture change BA cecal content composition. (**A**) Ratio Free/conjugated BA; (**B**) Ratio primary/secondary BA; (**C**) Different BA species expressed as percentage of Total BA. Results are expressed as mean ± SEM. # *p* < 0.05, ## *p* <0.01, HFD vs. LFD, * *p* < 0.05, ** *p* < 0.01, treated vs. HFD, Mann-Whitney test.

**Table 1 nutrients-13-00713-t001:** Bacterial strains used in this study with their origins.

Strains Designation	Bacterial Species	Origin
LA306	*Bifidobacterium animalis* subsp. *lactis* (*B. lactis*)	Human
LA803	*Bifidobacterium bifidum*	Human
LA804	*Bifidobacterium animalis* subsp. *lactis* (*B. lactis*)	Human
LA805	*Bifidobacterium breve*	Human
PI10	*Bifidobacterium longum* subsp. *longum* (*B. longum*)	Dairy product
PI22	*Bifidobacterium bifidum*	Human
LA305	*Lacticaseibacillus rhamnosus*	Human
LA307	*Ligilactobacillus salivarius*	Animal
LA806	*Lactobacillus gasseri*	Human
PI2	*Ligilactobacillus salivarius*	Unknown
PI3	*Lactiplanibacillus plantarum*	Vegetable
PI4	*Lactobacillus acidophilus*	Human
PI5	*Lactobacillus helveticus*	Dairy product
PI8	*Lacticaseibacillus casei*	Dairy product
PI11	*Lactobacillus acidophilus*	Human
PI13	*Lactobacillus helveticus*	Dairy product
PI15	*Lacticaseibacillus paracasei*	Human
PI18	*Lacticaseibacillus paracasei*	Human
PI19	*Lactiplantbacillus plantarum*	Dairy product
PI20	*Lacticaseibacillus casei*	Human
PI28	*Lacticaseibacillus rhamnosus*	Human
PI21	*Streptococcus thermophilus*	Dairy product
PI23	*Lactococcus lactis*	Unknown

## Data Availability

Data is contained within the article or Appendix A.

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
