# Peer review of "Multiple Selection Criteria for Probiotic Strains with High Potential for Obesity Management"

_nutrients, 2021, doi:10.3390/nu13030713_

Round 1

Reviewer 1 Report

The work presented in this paper is interesting and well presented.

Minor comments:

Figure 7: The author might consider changing the panel E-K from a histogram to a table.

Major comments:

  1. The authors showed that four bacterial strains have decreased lipid accumulation by Oil-red O quantifications. The authors should consider screening for the main adipogenic genes like adiponectin, PPARg, LPL and FABP4 to confirm the data.
  2. The authors mentioned that the bacterial strains have an impact on gut permeability. Did the authors look for the expression of junctional complexes (thigh junctions, adherens or gap junctions) in enteroendocrine cells?
  3. The authors tested the effect of bacterial strains on 3T3-L1 preadipocytes. Would the authors consider using human adipose-derived stem cells to test the effect of probiotics on lipid accumulation?
  4. Why the authors think that none of the probiotic strains has an effect on the in vivo tolerance to glucose?
  5. In Figure 5, the expression of inflammatory genes was evaluated in visceral adipose tissue. How about the SCAT? The authors might also consider increasing the panel of inflammatory makers to include IL-1b, IL-6, VEGF and TGFb.

Reviewer 2 Report

Using an obesity-induced mouse model, the authors found obesity-suppressing and inflammation-suppressing effects of certain bacteria, as well as effects on hormones that control energy balance. These effects were thought to be related to changes in the composition of short-chain fatty acids and bile acids in the intestine due to changes in the composition of intestinal bacteria. While this is well written and discussed the paper, I have some comments for the authors.

1. In the animal experiment, was the bacterium administered only for the first 5 days, or was it continued for 15 weeks until sacrifice?

2. Why was the expression of TGR5 studied in the duodenum and not in the distal small intestine? Do the authors assume that the administered probiotics act directly on the duodenum?

3. Mice on a high-fat diet are known to have constipation. Did constipation also occur in the present model? And was the constipation improved by the administration of probiotics?

4. Figure 7. uses a different color for the bar graph than the other figures. Since it is a little difficult to see, we suggest that the colors be aligned with the other graphs.
